# Simulation and Comprehensive Analysis of AlGaN/GaN HBT for High Voltage and High Current

**Xinyuan Wang** [1,2] , **Lian Zhang** [1] , **Jiaheng He** [1,2] , **Zhe Cheng** [1] , **Zhe Liu** [1,2] **and Yun Zhang** [1,2,*]

1. Laboratory of Solid State Optoelectronics Information Technology, Institute of Semiconductors, Chinese Academy of Sciences, Beijing 100083, China; wangxinyuan@semi.ac.cn (X.W.); zhanglian07@semi.ac.cn (L.Z.); hjh22@semi.ac.cn (J.H.); zhecheng@semi.ac.cn (Z.C.); liuzhe@semi.ac.cn (Z.L.)
2. Center of Materials Science and Optoelectronics Engineering, University of Chinese Academy of Sciences, Beijing 100049, China
* Correspondence: yzhang34@semi.ac.cn

**Abstract:** We present a series of TCAD analysis of gallium nitride (GaN) heterojunction bipolar transistors (HBTs) that investigates the impact of various key parameters on the gain characteristics, output characteristics, and breakdown characteristics. It has been observed that the DC gain of the AlGaN/GaN HBTs exhibits a non-linear relationship with the increase in the Al fraction. Specifically, the DC gain initially rises, then declines after reaching its peak value at approximately 7%. By optimizing the concentration of the base and the concentration and thickness of the collector epitaxial layer, it is possible to achieve devices with breakdown voltages of 1270 V (with a collector thickness of 6 μm and a carrier concentration of $2 \times 10^{16}$ cm$^{-3}$), specific on-resistance of 0.88 mΩ·cm$^2$, and a current gain of 73. In addition, an investigation on breakdown characteristics is conducted for HBTs with two types of substrates, namely QV-HBTs and FV-HBTs, at different inclinations of the ramp. We propose that critical angles are 79° and 69° to prevent the surface breakdown of the device, which helps to achieve an avalanche in GaN HBTs. We anticipate that the aforementioned findings will offer valuable insights for designing GaN-based power HBTs with elevated breakdown thresholds, heightened current densities, and increased power capabilities.

**Keywords:** GaN; heterojunction bipolar transistors (HBTs); TCAD; vertical power devices; breakdown voltage; current gain; beveled edge

## 1. Introduction

Vertical GaN transistors have exhibited significant potential for power switching purposes due to their ability to operate at higher voltages, accommodate higher current densities, and exhibit lower specific on-resistance ($R_{ON,sp}$) [1–3]. Recently, fin-channel junction field-effect transistors (Fin-JFET) with an avalanche breakdown voltage of 1200 V have been reported [4]. The utilization of sub-micrometer fin channels in Fin-JFETs facilitates the attainment of enhanced gate control and the establishment of a normally off operational state. On the other hand, GaN heterojunction bipolar transistors (HBTs), which also possess vertical structures have also been proposed as a promising alternative for power switching [5–8] with the advantages of low photolithography accuracy requirement, normally off operations, high current density, strong avalanche breakdown ability, and lower $R_{ON,sp}$ due to conductivity modulation effect. Bipolar transistors have been widely used in the SiC material system (SiC BJT), while the development of GaN bipolar transistors has lagged behind due to the material constraints of the basic difficulties. However, up to now, several encouraging findings have been documented regarding GaN HBTs, including the observation of a high electric field near 3 MV/cm [7–9] and a high current density of 141 kA/cm$^2$ on GaN-on-GaN HBT [10].

Despite the potential demonstrated by AlGaN/GaN HBTs as power devices, the fabrication process for GaN-based HBTs with high current gain ($\beta$) and high power encounters

various challenges, including epitaxial growth of high quality, thicker collectors, and epitaxial growth of p-type substrates with high carrier lifetimes and high hole concentrations for n-p-n HBTs [9]. The PND of GaN-on-Si at 1 kV level and the PND of GaN-on-GaN at 6 kV have been gradually reported [11,12] as the epitaxial technology of GaN materials has improved. It is anticipated that GaN HBTs operating at a voltage level of 1 kV can be fabricated utilizing the epitaxial structure. It is imperative to conduct structural validation and analysis of the electrical characteristics of GaN HBTs via TCAD simulation. However, there is currently a lack of comprehensive research on GaN HBTs utilizing TCAD. Ref. [9] describes the electrical characteristics of GaN/InGaN HBTs with composition-graded p-InGaN base; however, it does not delve into a comprehensive investigation of the impact of epitaxial layer design on the gain, output, and breakdown characteristics of GaN HBTs.

Hence, this paper undertakes a comprehensive structural simulation of AlGaN/GaN HBTs. The forward conduction characteristics of HBTs are investigated, including the effect of Al composition on the conduction characteristics and the current distribution in the turn-on state. The reverse blocking characteristics of HBTs are investigated, including avalanche breakdown of HBTs, the dependence of different base-collector epitaxial layer designs on breakdown voltage, and the effect of beveled edge on high-voltage HBTs. Finally, a GaN HBT with a breakdown voltage of 1270 V is designed with a collector thickness of 6 μm and a carrier concentration of $2 \times 10^{16}$ cm$^{-3}$. The Fully Vertical HBT (FV-HBT) and Quasi-Vertical HBT (QV-HBT) are compared and analyzed, and their specific on-resistance is deduced to be 0.88 mΩ·cm$^2$ and 1.02 mΩ·cm$^2$, with a maximum current gain of 56 and 73, respectively, which is inferred from the simulation results. The results provide theoretical guidance for the development of high-performance GaN-based HBTs, whose potential applications include photovoltaic inverters, data servers, and electric vehicles (EVs), where GaN HBTs can be used as traction inverters, for example.

## 2. Simulation Details

The simulation tool used in this paper to build the AlGaN/GaN HBT model and perform the numerical calculations is implemented in TCAD. The simulation incorporates various physical models. Three main equations are included first: the Poisson equation, the carrier diffusion equation, and the continuity equation. In addition, there are high-field-velocity saturation models, carrier generation-recombination models, and low-field mobility models. The main model used to simulate device breakdown is the impact ionization model, which can be specifically described using the Chynoweth equations. From the literature, it can be derived that the impact ionization coefficients of the GaN material $\alpha_n$ and $\alpha_p$ change as a function of the applied electric field [13–15], and the formulae are as follows:

$$\alpha_e(E) = 2.90 \times 10^8 cm^{-1} \cdot \exp\left(-\frac{3.40 \times 10^7 V/cm}{E}\right)$$

$$\alpha_h(E) = 1.34 \times 10^8 cm^{-1} \cdot \exp\left(-\frac{2.03 \times 10^7 V/cm}{E}\right)$$

Combined with Chynoweth's equations, it is possible to obtain the impact ionization generation rate (G), and the integral of G can thus analyze the location where the avalanche breakdown occurs. In addition to this, an incomplete ionization model is employed to reflect the ionization process of the ionized acceptor (Mg) within GaN under equilibrium conditions. It is observed that the acceptor energy level of Mg is 150 meV above the valence band. Consequently, the concentration of ionized holes in GaN can be calculated by the following equation [16]:

$$N_A^- = \frac{N_A}{1 + g_A \exp\left(\frac{E_A - E_F}{kT}\right)}$$

where the acceptor degeneracy factor, denoted as $g_A$, is assigned a value of 4; $E_F$ is the Fermi energy level, and $E_A$ is the acceptor energy level; $N_A$ is the doping concentration

of Mg impurity, and the concentration was set as $3 \times 10^{18}$ cm$^{-3}$ for the *p*-GaN base layer. The carrier complexation mechanism is described by the SRH and Auger models, and the lifetime of the minority carriers (electrons) is uniformly set to 1 ns in the base. The activation of the polarization effect model is required due to the polarization effect that occurs at the AlGaN/GaN interface.

In order to establish the fidelity of the model and parameters employed in this study, a calibration process was conducted by comparing it with the I-V characteristics obtained from the p-i-n diode via experimental means [17]. Figure 1a compares the forward and reverse I-V characteristics, and the better agreement between the two demonstrates the accuracy of the model.

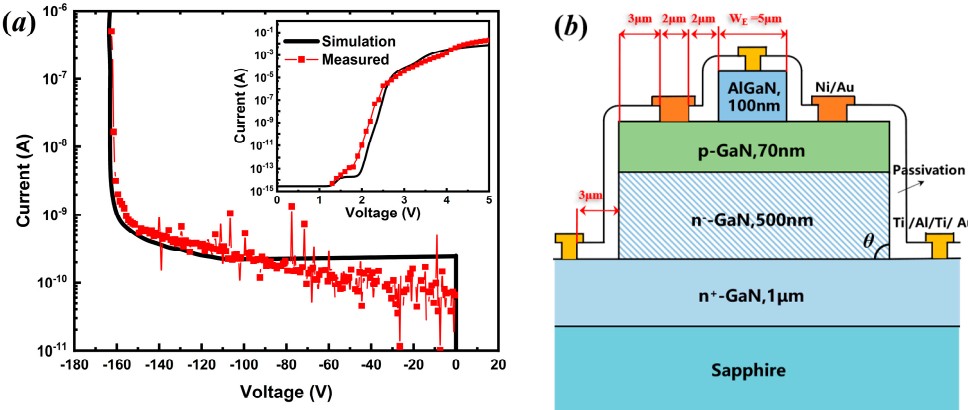

**Figure 1.** (**a**) Forward bias (inset) and reverse breakdown comparison of the simulated and measured data; (**b**) the cross-section structure of the AlGaN/GaN HBT.

The structure of the HBT device is shown in Figure 1b, which is the same as the HBT device in the literature [7]. The thickness and doping concentration of each layer are detailed in Table 1, and the dimensions of the device are defined from top to bottom as follows: emitter length $W_E$ = 5 μm, base length $L_B$ = 19 μm. The electrode dimensions are uniformly set to 2 μm, and the relative positions are as shown in the figure, and the angle between the BC mesa and the horizontal is defined as θ, which is set to 90° for all the places not mentioned below. In order to ensure accuracy and avoid any potential asymmetry in the calculated results, the simulation was conducted on only one-half of the device, taking into account its inherent symmetry.

**Table 1.** AlGaN/GaN HBT layer structure.

| Region | Material | Thickness (nm) | Doping (cm$^{-3}$) | $\mu$ (cm$^2$/V·s) |
|---|---|---|---|---|
| Emitter | AlGaN | 150 | [Si] = $3 \times 10^{18}$ cm$^{-3}$ | 290 |
| Base | GaN | 70 | [Mg] = $3 \times 10^{18}$ cm$^{-3}$ | 14 |
| Collector | GaN | 500 | [e$^-$] = $1 \times 10^{17}$ cm$^{-3}$ | 400 |
| sub-Collector | GaN | 1000 | [Si] = $3 \times 10^{18}$ cm$^{-3}$ | 150 |

Much effort has been made to reduce the complexity of the model and obtain good results using mathematical methods [18–22]. However, in three dimensions, the motion of the carriers is more complex, so performing the calculations still requires a lot of computational effort. Since the ideal GaN HBT model is homogeneous along the *z*-axis, we simplify it to a 2D model, which reduces the distribution data on the *z*-axis. The 2D model is able to satisfy the needs of our simulation, considering that we focus on the effects of epitaxial layer doping, thickness, and angle θ on the electrical performance.

## 3. Result and Discussion

### 3.1. Impact of Al Mole Fraction of Emitter on the Forward Performances of GaN HBTs

The Al mole fraction in AlGaN emitters typically ranges from 5% to 20% as reported in AlGaN/GaN HBTs [7–9,23,24]. However, it is widely recognized that the crystal mass quality in $Al_xGaN$ deteriorates as the Al mole fraction increases, as deep-level increments and deep donor formations become conspicuous [25]. Therefore, considering practical implications, the Al content of the emitter is reduced with the aim of enhancing the performance and consistency of the HBT. Hence, there is significant interest in establishing a minimum threshold for the Al component in HBTs. The following section provides empirical findings regarding the influence of emitter Al content on current gain in AlGaN/GaN HBTs.

As shown in Figure 2a, $\beta$ initially increases as the Al composition increases, attains its maximum value at approximately 7% Al composition, and subsequently decreases. This trend aligns with the behavior of $I_C$ in Figure 2b, whereas $I_B$ continues to decrease with increasing $x$. The phenomenon can be explained by the energy band structure of the AlGaN/GaN HBT, as shown in Figure 2c.

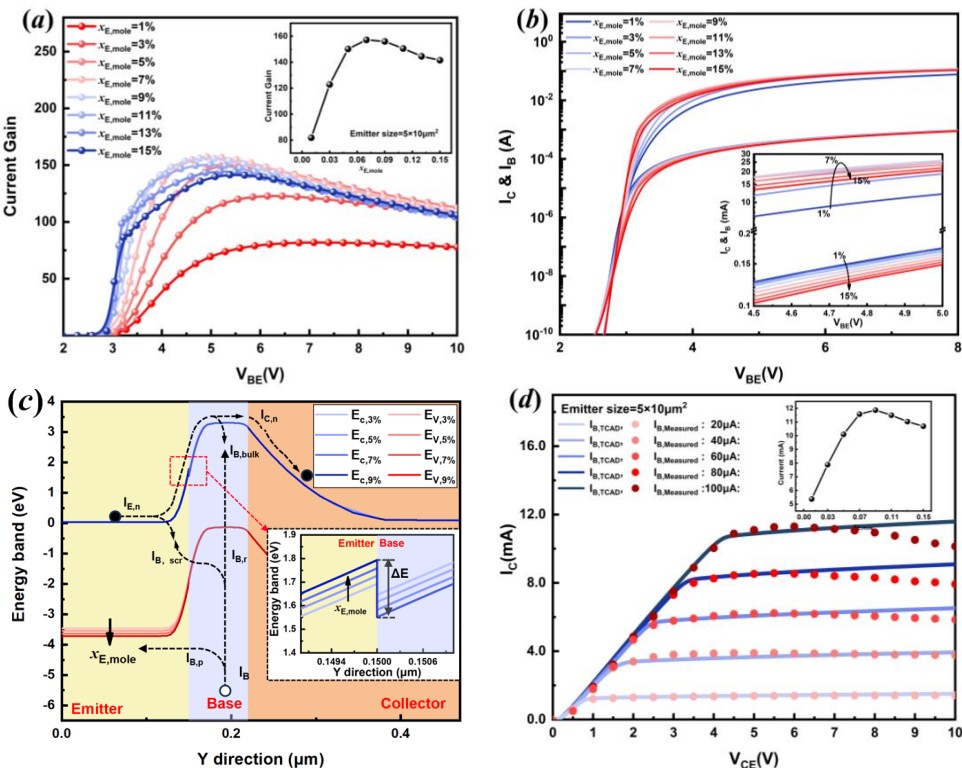

**Figure 2.** The simulated (**a**) current gain plots (Inset: maximum current gain as a function of Al fraction) and (**b**) $I_C$ & $I_B$ of HBTs with different Al fraction; (**c**) energy band lineup of an $Al_xGa_{1-x}N$/GaN HBT; (**d**) the cross-section structure of the AlGaN/GaN HBT. Inset: maximum saturation current as a function of Al fraction.

As shown in Figure 2c, the energy band diagram and current composition of the HBT are demonstrated. $I_{nE}$ and $I_{pE}$ are the electron diffusion currents injected from the emitter region to the base region and hole diffusion currents injected into the emitter from the base, respectively. $I_{B,scr}$ and $I_{B,bulk}$ are denoted as the complex currents in the space charge region of the emitter junction and the complex currents of the body in the base region, respectively, along with $I_{B,bulk}$, collectively constitute the base region current $I_B$. Thus, the DC current gain can be expressed as follows:

$$\beta = \frac{I_C}{I_B} = \frac{I_{E,n} - I_{B,bulk} - I_{B,scr}}{I_{B,P} + I_{B,bulk} + I_{B,scr}}$$

The rise in the energy band offset $q\Delta E_V$ of the BE junction leads to a decrease in the backward injection current $I_{B,p}$ from the base into the emitter. Consequently, the $I_B$ decreases with the increase in $x$, leading to a gradual increase in $\beta$, as shown in Figure 2b. As $x$ continues to increase, $\beta$ starts to decrease due to the creation of a potential well $\Delta E$ at the interface between emitter and base as shown in Figure 2c, which collects the injected electrons and increases the complex losses, thus decreasing $I_{E,n}/I_{B,p}$, and hence $\beta$. And $\Delta E$ increases with $x$ so that $\beta$ and $I_C$ reach a maximum value at $x = 7\%$ and then decreases gradually. At an $x$ value of 7%, the maximum $\beta$ value is observed to be 157.

Figure 2d shows the output curve of the modeled HBT at $I_B$ from 20 μA to 100 μA, which shows a better fit of the simulation results to the measurements in [7]. It is worth noting that in the simulated curves, due to the Early effect, $I_C$ rises slowly with increasing $V_{CE}$ when $V_{CE}$ reaches $V_{knee}$ [9]. However, in the test curve, at larger $I_B$ (at 60/80/100 μA), $I_C$ decreases with increasing $V_{CE}$ due to the self-heating effect. The inset shows the maximum saturation current as a function of the Al fraction, again showing an increasing and then decreasing trend.

The current gain of the HBT is inversely proportional to the width of the base region; a narrower base helps to reduce the $I_{B,bulk}$ and hence increase the $\beta$. However, the electric field needs to drop to 0 in a base of sufficient thickness to prevent the base from penetrating, and hence the base thickness cannot fall below a certain minimum value. As shown in Figure 3a, the current gain gradually decreases to almost 0 as the thickness of the base increases, and in order to obtain a current gain of not less than 150, the thickness of the base should not be higher than 0.07 μm. In addition, $\beta$ can likewise be increased by improving the quality of the base material and thus the minority carrier lifetime of the base, as shown in the following equation:

$$\frac{1}{\beta} = \frac{\tau_B}{\tau_e} + \frac{x_B N_B D_{pE}}{x_E N_E D_{nB}} \exp\left(-\frac{\Delta E_V}{k_B T}\right)$$

where $\tau_e$ is the minority carrier lifetime in base, $\tau_B$ is the base transport time, $x_B$ and $x_E$ are the thicknesses of the base and emitter, respectively, $N_B$ and $N_E$ are the respective carrier concentrations, $D_{pE}$ and $D_{nB}$ are the diffusion coefficients of holes and electrons in the emitter and base, respectively [26]. Figure 3b demonstrates the impact of the minority carrier lifetime on the current gain. It is observed that the current gain exhibits an upward trend as the minority carrier lifetime increases; however, the rate of growth slows down at minority carrier lifetimes greater than 2 ns. According to the data in the figure, for the AlGaN/GaN HBT to obtain a current gain greater than 150, the minority carrier lifetime should be no less than 1 ns.

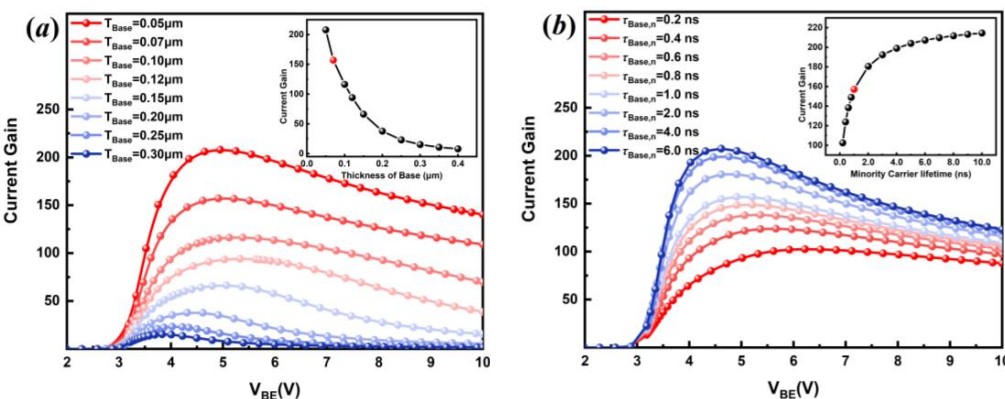

**Figure 3.** (**a**) Current gain of HBT with different base thicknesses as a function of $V_{BE}$. Inset: Maximum current gain as a function of Base thickness; (**b**) Current gain of HBT with different minority carrier lifetimes as a function of $V_{BE}$. Inset: Maximum current gain as a function of minority carrier lifetime.

### 3.2. Forward (and Impact of Device Structure on It) and Reverse Blocking Performances of GaN HBTs

Heterogeneous substrate GaN technology (Si-based, Sapphire-based) is a technique for growing epitaxial layers of GaN on inexpensive substrates. GaN *p*-i-*n* diodes (PND) and Schottky barrier diodes (SBDs) are highly competitive in the marketplace due to the gradual maturation of heterogeneous growth techniques and the ability to meet performance requirements at low costs, achieving breakdown voltages over 1 kV and low on-resistance [12,27], and GaN-on-Si transverse power devices up to 650 V have been successfully commercialized [28]. Vertical GaN power devices grown on homogeneous substrates (GaN-on-GaN) are preferred because of their lower thermal and lattice mismatch. GaN vertical power devices have a significant advantage in achieving high power density, thermal management, and low surface trap sensitivity because the electric field (E-field) is far from the device surface. The *BV* can be increased by increasing the thickness of the voltage-blocking layer, while the $R_{ON}$ increases only slightly due to external phenomena such as conductivity modulation. The peak electric field of vertical devices is far inside the GaN layer, and avalanche breakdown is present, thus improving device reliability and eliminating the need to overdesign devices [29,30].

The available literature describes two configurations of AlGaN/GaN structures. The first configuration is known as the quasi-vertical heterojunction bipolar transistor (QV-HBT), where the collector and emitter base are positioned on the same side, resulting in a quasi-vertical arrangement. The second configuration is referred to as the fully vertical heterojunction bipolar transistor (FV-HBT), where the collector and emitter base are located on opposite sides, forming a completely perpendicular structure. Figure 4a shows the cross-section of the collector current density of the two structured HBTs in the turn-on state, when the $I_B$ and the applied collector voltage $V_{CE}$ are 100 μA and 10 V, and the maximum saturation current of the qV-HBT in this state is 13.2 mA, while that of the FV-HBT is 61.8 mA, as shown in Figure 4b. It is evident that a significant portion of the base and collector regions under the base metal exhibit negligible current flow. The collector area experiences a lack of current flow, rendering it an unutilized region. The comparison between FV-HBT and QV-HBT reveals that the primary region of current conduction is directly below the emitter area of FV-HBT, which has a more uniform current distribution compared to QV-HBT, while the QV-HBT device has transverse current conduction due to the fact that the emitter area and the collector area are located on the same side The current of the emitter area is therefore concentrated in the emitter electrode close to the side of the collector area. In the figure below, the direction of current flow in the turn-on state is shown with the emitter region as the endpoint. It can be seen by the direction of the current flow.

Intercepting the current distribution in the center area of the emitter and collector regions, respectively, to obtain the lateral distribution of the current in each layer is shown in Figure 4c. From the figure, it can be seen that in the vicinity of the emitter electrode (at $x = \pm 2.0$ μm), the $I_E$ and $I_C$ of the FV-HBT ($1.11 \times 10^6$ A/cm$^2$, $3.82 \times 10^5$ A/cm$^2$) are significantly higher than those of the QV-HBT ($4.33 \times 10^5$ A/cm$^2$, $1.29 \times 10^5$ A/cm$^2$), and there is still a higher current density in the center region of the emitter area, which demonstrates its whole emitter area conduction current advantage. The current distribution graph in Figure 4c shows that the $I_C$ decreases from $3.82 \times 10^5$ A/cm$^2$ to $1.03 \times 10^5$ A/cm$^2$ from the edge of the electrode to the central region in the FV-HBT. In contrast, the $I_C$ of the QV-HBT decreases from $1.29 \times 10^5$/cm$^2$ to almost zero ($3.01 \times 10^3$). For the quasi-vertical structured HBTs, the calculation of their specific conduction resistances, $R_{ON,sp}$, or the current densities cannot be conducted simply using the area of the emitting area or the area of the collector area to calculate. In addition, HBTs are potential power electronic devices due to their high current density, strong avalanche breakdown capability, and low specific on-resistance, where the breakdown voltage can be increased by increasing the thickness of the drift region, and scaling high on-current can be achieved by increasing the lateral size of the device. However, many current reports on GaN-based HBTs are based on sapphire

substrates, and the heterogeneous growth of GaN not only makes it challenging to obtain a larger thickness of the drift region but also fails to give full play to its potential for scaling current.

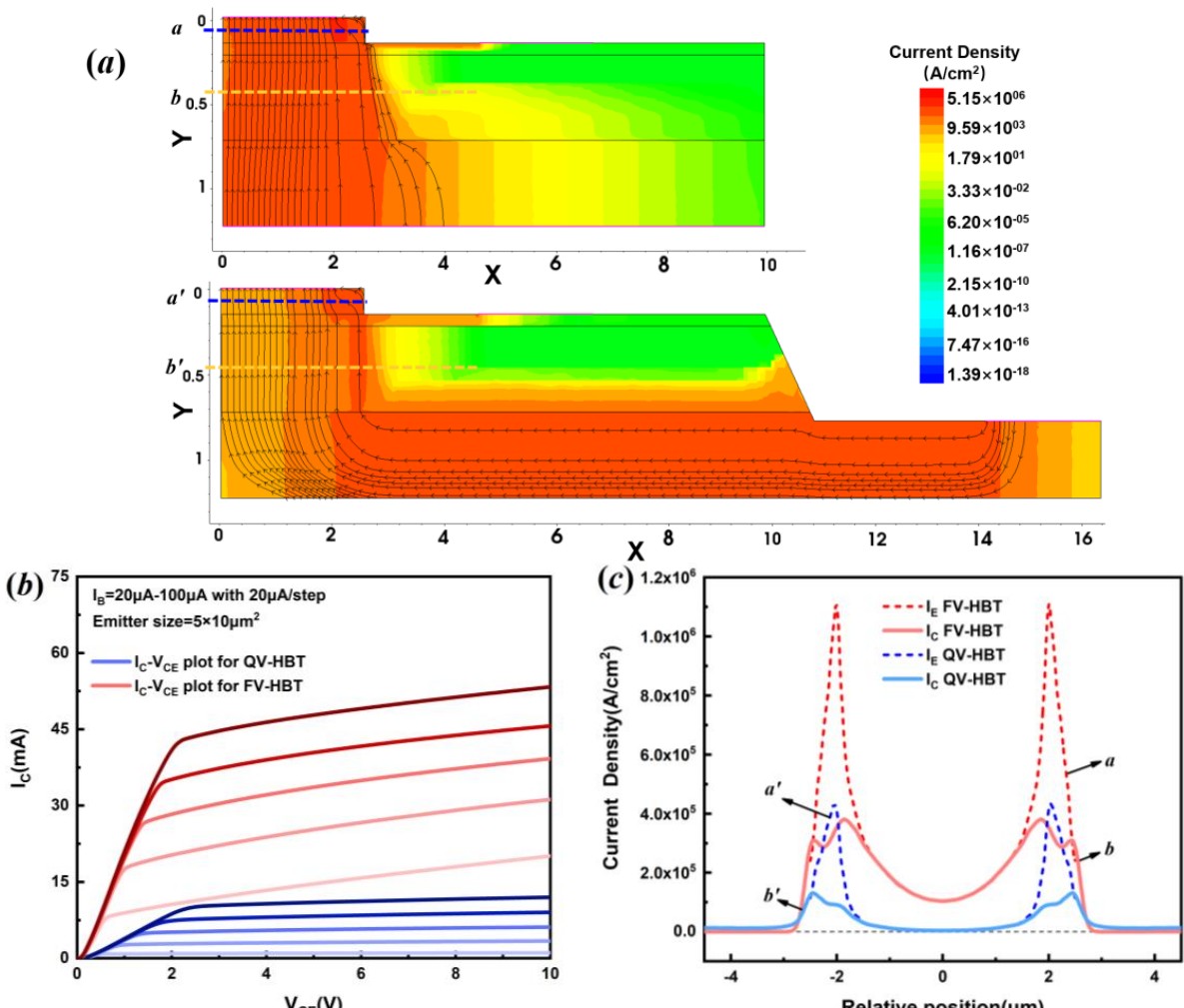

**Figure 4.** (**a**) Schematic current distribution of FV-HBT and QV-HBT in on-state. The dotted line in the figure shows the center of the emitter (*a* and *a′*) and the collector (*b* and *b′*); (**b**) output curves at the same $I_B$ for FV-HBT and QV-HBT; (**c**) distribution of current density along the dashed line in (**a**) with respect to mutual position.

Figure 5a shows the breakdown characteristic curve of the same HBT structure reported in [7]. An avalanche breakdown voltage of 165 V (@$I_{CEO}$ = 4 nA) was obtained, which agrees well with the experimental data, and the inset shows the distribution curve of the electric field strength along the cross-section. As shown in Figure 5b–d, the electric field distribution, current distribution, and impact ionization distribution inside the HBT when the breakdown voltage is reached are simulated based on the HBT structure in [7]. As shown in Figure 5b, when the emitter is grounded, and the collector is connected to a high voltage, the voltage mainly falls on the low-doped collector, and avalanche breakdown is an impact ionization (I.I.) and multiplication process, which usually occurs at the p-n junction [31]. Since the current conduction path of HBT avalanche breakdown is collected from the emitter through the base and the collector directly below the emitter and finally converges to the collector, there is almost no lateral current diffusion in the collector, as shown in Figure 5c. Hence, the I.I. mainly occurs at the BC junction directly underneath the emitter, as shown in Figure 5d, and the I.I. generated electrons are pushed towards the

collector, while a large number of electrons are extracted from the emitter to recombine with the I.I. generated holes.

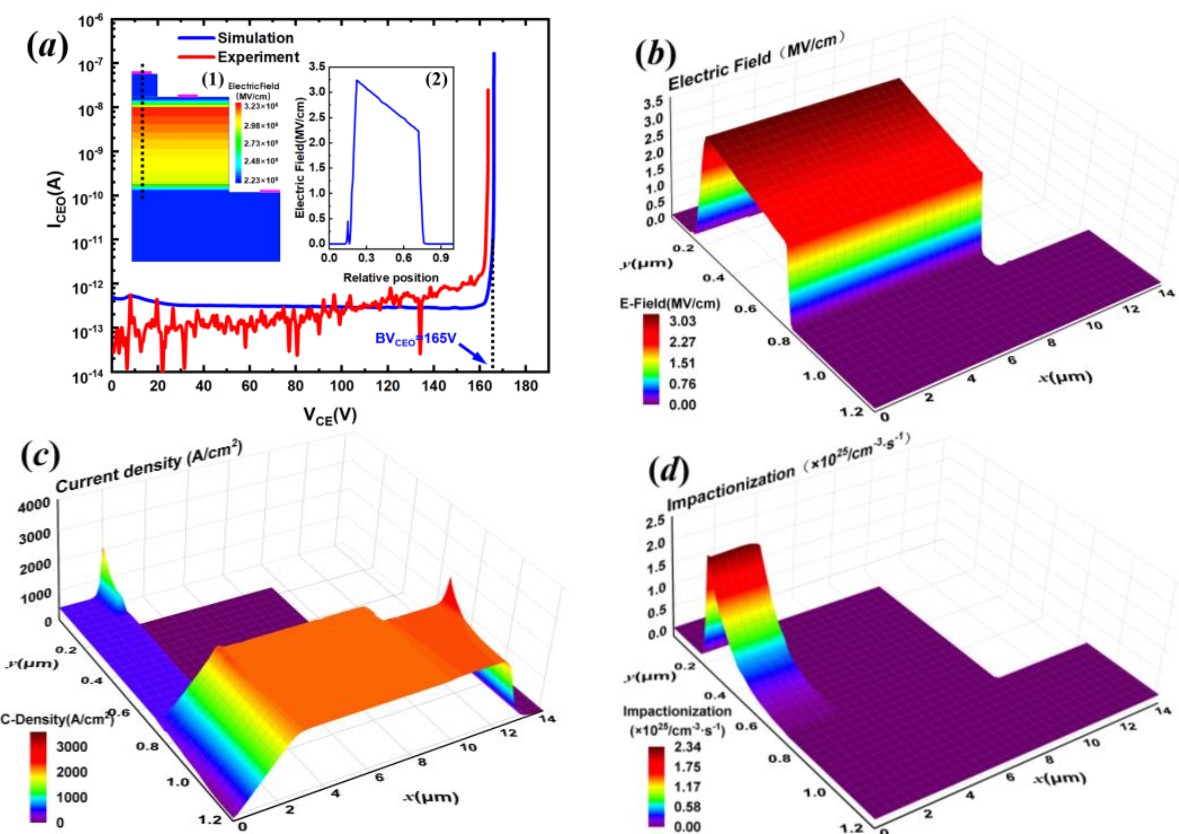

**Figure 5.** (**a**) Image of breakdown voltage with the same epitaxial structure as in article [7], inset (1) shows the device structure, and the inset (2) shows the distribution of the electric field strength along the dashed line in the inset (1) with the relative position of the HBT. (**b**) Schematic of the electric field strength; (**c**) schematic of the current distribution; (**d**) schematic of the I.I. distribution along the dashed line in Figure (**a1**) under avalanche breakdown.

GaN devices are often etched to form a beveled angle during preparation, with measured beveled angles between 40° and 90° (depending on the type of mask used for etching) [32,33], and this phenomenon is also present in the base-collector (BC) mesa of GaN HBTs. The reduction in the BC mesa beveled angle also results in a lower breakdown voltage, which will be discussed in Section 3.3.

### 3.3. Impact of Epitaxial Layer Design and BC Mesa Bevel on the Blocking Performances Characteristics of GaN HBTs

As mentioned before, GaN-based HBTs have great potential as power electronic devices. Many reports have demonstrated the high current density and low on-resistance of HBTs; however, more verification has yet to be conducted for the high-voltage performance of GaN HBTs, and only a breakdown voltage of 330 V has been achieved [5].With the enhancement of GaN epitaxial technology, the HBTs with thicker drift layers contribute to the avalanche higher breakdown, so device simulation for GaN HBT devices with high breakdown voltage is vital. Figure 6a shows the breakdown voltages of two types of HBTs (FV-HBT and QV-HBT, solid lines and points, respectively) at a collector concentration of $2 \times 10^{16}$ cm$^{-3}$ and a collector thickness of 0.5/1.0/2.0/3.0/4.0/6.0. From Figure 6a, it can be seen that the QV-HBT has the same avalanche breakdown voltage as the FV-HBT under the ideal conditions of a perfectly vertical BC mesa, and the breakdown voltages that can be achieved are 180/332/594/826/1012/1272 V, so that a breakdown voltage of 1.2 kV level

can be achieved with a collector of 6 μm. The inset shows the electric field distribution of the base and collector when the breakdown occurs, and it can be seen by the inset that the different thicknesses of the collector mentioned above are Punch-through (PT) conditions. The thickness of the collector over 6 μm at this concentration will not significantly increase the breakdown voltage.

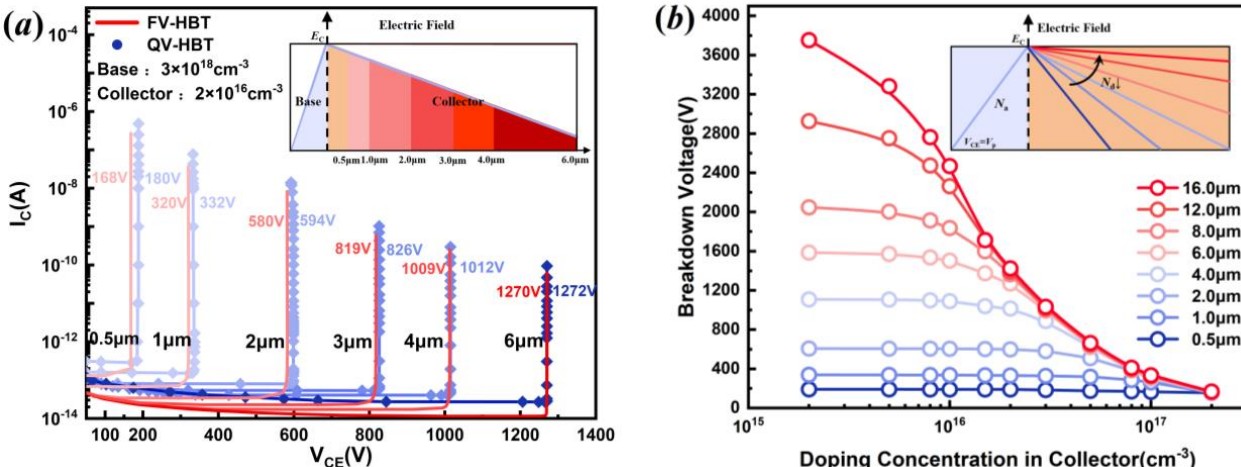

**Figure 6.** (**a**) Image of the breakdown voltage of QV-HBT and FV-HBT as a function of collector thickness; (**b**) breakdown voltage of HBT with different collector lengths as a function of n-doping concentrations of the collector.

Figure 6b demonstrates the breakdown voltage as a function of doping concentration for different thicknesses of the collector. Since decreasing the doping concentration can slow down the rate of the electric field strength decreasing in the space charge region, the BV increases with decreasing doping concentration and gradually reaches saturation when it is reduced to a certain concentration. The larger the thickness of the collector is, the lower the concentration that reaches saturation is. This is because decreasing the collector concentration can change from the Non-Punch-through (NPT) state to the PT state, and when the carrier concentration is small enough, continuing to decrease the carrier concentration will not yield as much as increasing the collector thickness to increase the breakdown voltage.

Figure 7a demonstrates the breakdown voltage for different base doping concentrations under the same collector condition (6 μm, $2 \times 10^{16}$ cm$^{-3}$). Figure 7a shows the effect of $p$-GaN doping concentration on the breakdown characteristics of the collector region for different background doping concentrations. It can be seen from Figure 7a that the doping concentration of the $p$-GaN has a significant effect on the breakdown characteristics of the HBTs, and it is essential to point out that the doping concentration is the concentration of magnesium dopant. As the $p$-GaN doping concentration decreases, the $BV$ of the HBT first remains constant and then starts to decrease after reaching a certain concentration ($2.7 \times 10^{18}$ cm$^{-3}$ for collector doping is $2 \times 10^{16}$ cm$^{-3}$) and eventually falls below 100 V. This is due to the base punch-through effect; when the reverse voltage on the collector junction increases, the collector depletion layer expands to both sides, and the $W_B$ decreases accordingly. When the collector junction reverse bias reaches a certain value $V_{pt}$, although the collector junction has not yet avalanche breakdown, the single $W_B$ has already decreased to 0, and at this time, the base becomes a depletion region completely, and the base region depletion layer can be solved by the base depletion layer formula for the $V_{pt}$, as the following equation:

$$W_B = \left[ \frac{2\varepsilon_s N_C}{q N_B (N_C + N_B)} (V_{bi} + V_{pt}) \right]$$

where $V_{bi}$ is the built-in potential, and $V_{pt}$ can be solved using the width of the depletion region on the base side equal to $W_B$, and the following equation neglects $V_{bi}$,

$$V_{pt} = \frac{qN_B(N_C + N_B)}{2\varepsilon_s N_C}W_B^2$$

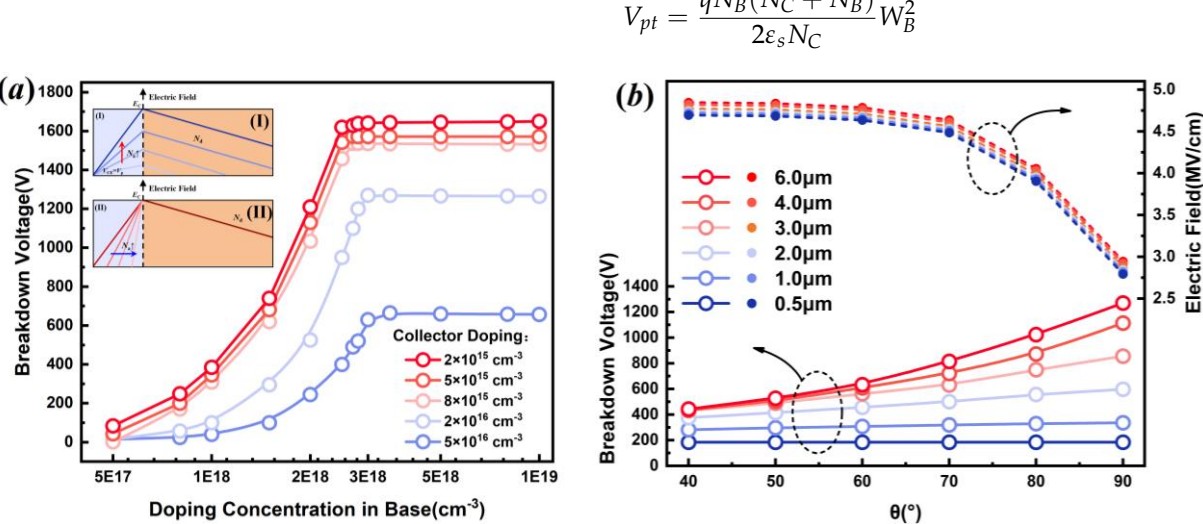

**Figure 7.** (**a**) Breakdown voltage of HBT with different collector doping as a function of p-doping concentrations of the base; (**b**) breakdown voltage and electric field of HBT with different collector lengths as a function of BC mesa bevel angles.

When $V_{CE} = V_{PT}$, $V_{CE}$ continues to increase to make the BE junction open, and there will be a large number of emitter carriers injected into the punch-through base and then reach the collector, making $I_{CEO}$ increase dramatically, at this time according to the definition of the $BV_{CEO}$:

$$BV_{CEO} = V_{pt} + V_{F,BE} \approx V_{pt}$$

When the carrier concentration is low, base penetration occurs, and a higher reverse voltage can be borne on the collector as the doped Na increases, and thus the HBT has a higher breakdown voltage, as shown in inset (I) in Figure 7a. Furthermore, when Na is high enough, since the BC junction has already reached the peak electric field, continuing to increase Na will not cause the breakdown voltage to continue to increase, as shown in inset (II). When the *p*-GaN doping concentration is $2.8 \times 10^{18}$ cm$^{-3}$, the peak *BV* of the HBT with a drift doping concentration of $2 \times 10^{16}$ cm$^{-3}$ is 1234 V. When the *p*-doping concentration is further increased, the *BV* value remains unchanged. Nevertheless, it is worth noting that a significant increase in the doping concentration of the base has the adverse effect of diminishing the carrier lifetime within the base region, consequently leading to a reduction in the current gain. Consequently, the doping concentration of the base in the range of $(2.8–3.0) \times 10^{18}$ cm$^{-3}$ is the optimum doping concentration for obtaining a 1.2 kV HBT.

Since the sidewalls of GaN power devices exhibit sloped surfaces when performing mesa etching or device isolation, it will be a positive bevel if more material is removed in the lightly doped region, and it will be a negative bevel if more material is removed in the heavily doped region. For positive bevel, there is an area difference between the *p*-type material left behind and the *n*-type material. The depletion region on the lightly doped side has to be expanded inwardly to achieve the charge equilibrium, thus obtaining a surface E-field strength that is even lower than that of a planar junction with the same dopant. In the negative bevel edge termination, on the other hand, the lightly doped material leaves a large portion of it behind. This allows the depletion layer to reduce its area to balance the reduced charge in the heavily doped region, and yields a surface E-field strength higher than that of an identically doped planar junction. Consequently, the

positive bevel configuration is preferable for terminating the single high-voltage junction within high-voltage power rectifiers [34,35].

In the case of GaN, it is commonly observed that activating the *p*-type layer, which is buried beneath the *n*-type layer, poses challenges. As a result, the *p*-type layer is typically positioned on the top surface. This arrangement allows for the removal of the heavily doped *p*-type material during the top etching process, resulting in the formation of a negative bevel edge termination. However, this configuration leads to a reduction in the achievable breakdown voltage of the device.

Figure 7b demonstrates the images of breakdown voltage and maximum electric field strength variations at different BC mesa beveled angles for QV-HBT, for example. In the case of 1.2 kV HBT, the mesa beveled angles decrease from 90° to 40°, the breakdown voltage decreases from 1270 V to 420 V, and the peak field strength increases from 2.79 MV/cm to 4.69 MV/cm. The 2D electric field distributions of the simulated HBTs with different beveled angles $\theta$ of the slopes are shown in Figure 8a–f. In order to better show the trend and where the peak electric field occurs, the pictures show only a part of the device. Figures 8a and 9a demonstrate that the electric field is uniformly distributed in the HBT at a beveled angle of 90°, and the I.I. occurs predominantly in the region directly below the emitter. Observation of Figures 8b and 9b shows that when the beveled angle is less than 90°, the peak electric field appears at the edge of the BC mesa due to the bending of the space charge region. A second current path other than the region directly below the emitter appears along the sidewall surface of the HBT due to the contraction of the depletion layer, i.e., the current follows the outer surface of the BC mesa, flows transversely through the base and finally into the emitter, and I.I. starts to appear under the effect of the peak electric field at the edge of the BC mesa surface. It can be seen from Figures 8c–f and 9c–f that the curvature of the space charge region increases significantly with decreasing $\theta$, and its width decreases gradually. The width of the undepleted region with relatively higher concentration increases gradually, and the edge region of the mesa surface is larger than the center region, so the current in the second pathway increases gradually. The two currents are connected in parallel to form the leakage current ($I_{CEO}$) of the HBT, and as $\theta$ decreases, the peak of I.I. is transferred from the pn junction underneath the emitter (90°) to the pn junction at the edge of the BC mesa top (70°). The breakdown voltage decreases gradually in the process, which further decreases the depletion layer's width.

Simulation of a mesa surface with an angle between 70° and 80° (not shown in the figure) shows that the critical angle for the transfer of the maximum I.I from the central region to the mesa surface is 79°. As a result, when the BC mesa beveled angle generated during etching is lower than this angle, avalanche breakdown occurs at the mesa edge thus breaking down prematurely.

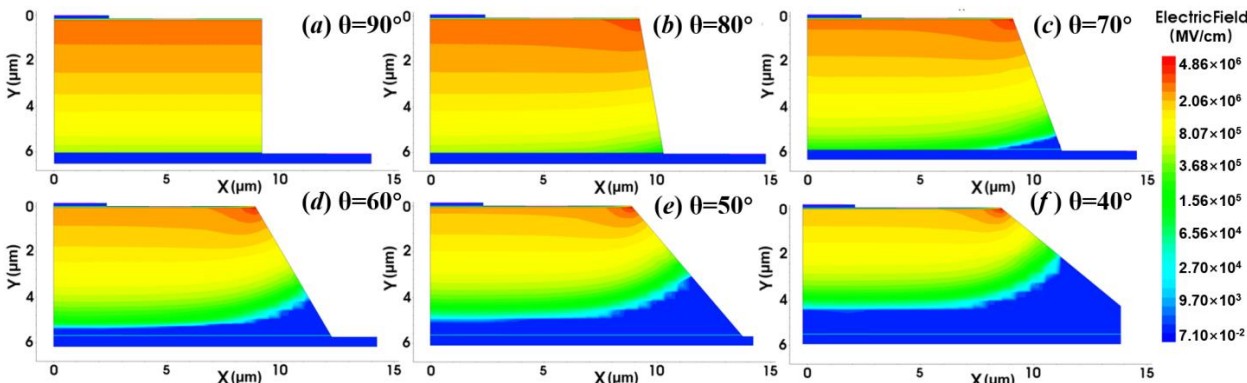

**Figure 8.** The 2D profile of electric field in the QV-GaN with $\theta$ = (**a**) 90°, (**b**) 80°, (**c**) 70°, (**d**) 60°, (**e**) 50°, and (**f**) 40°. It can be seen that the beveled angle decreases, and the electric field gradually concentrates at the edge of the BC mesa.

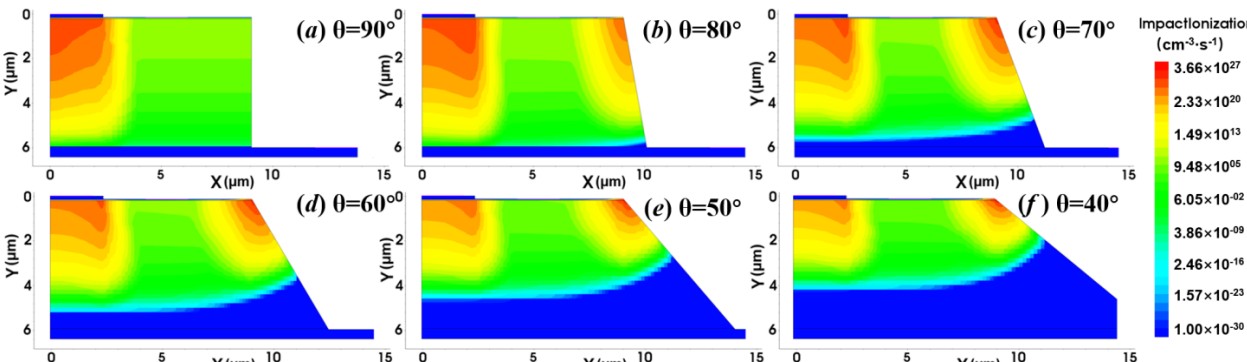

**Figure 9.** The 2D profile of I.I. in the QV-HBT with θ = (**a**) 90°, (**b**) 80°, (**c**) 70°, (**d**) 60°, (**e**) 50°, and (**f**) 40°. It can be seen that as the beveled angle decreases, when avalanche breakdown occurs, a conductive channel appears on the BC mesa surface and the I.I. maximum gradually shifts from below the emitter (80°) to the edge of the BC mesa surface (70°).

It is worth noting that in GaN power PNDs, when the inclination of the inclined plane is below a certain value, the location of the peak field strength gradually moves from the surface into the body, so continuing to reduce the inclination of the inclined plane will result in a lower peak field strength than the planar peak field strength, thus acting as an edge termination. This increases the breakdown voltage of the device, and in SiC BJT power devices, there are many reports on the use of multistep etch bases to form junction termination extensions (JTEs) to achieve higher breakdown voltages. However, in GaN, the process is more challenging to implement due to the severe damage and difficulty in recovering the *p*-GaN material after etching. Therefore, this paper will not do an in-depth related study.

For FV-HBT, Figure 10 illustrates the trend of electric field with decreasing beveled angle, which is similar to Figure 8, for angles less than 90°, a concentration of electric field appears at the edge of the BC mesa. Figure 11 demonstrates the trend of I.I. with decreasing beveled angle, the position of the maximum value of I.I. is at the center of the emitter electrode when θ is 90° (as shown in Figure 11a), and as θ decreases, the maximum value of I.I. moves closer to the edge of the emitter electrode and shifts to the edge of the emitter electrode at 70°, and finally when θ decreases to 60° (as shown in Figure 11d), the position of the maximum value of I.I. shifts to the surface of the BC mesa. Simulation of a mesa surface with an angle between 60° and 70° (not shown in the figure) yielded a critical angle of 69° for the occurrence of I.I. transfer from the central region to the mesa surface. As a result, the second current path of the FV-HBT dominates at 69°, and it is less prone to premature breakdown than the QV-HBT.

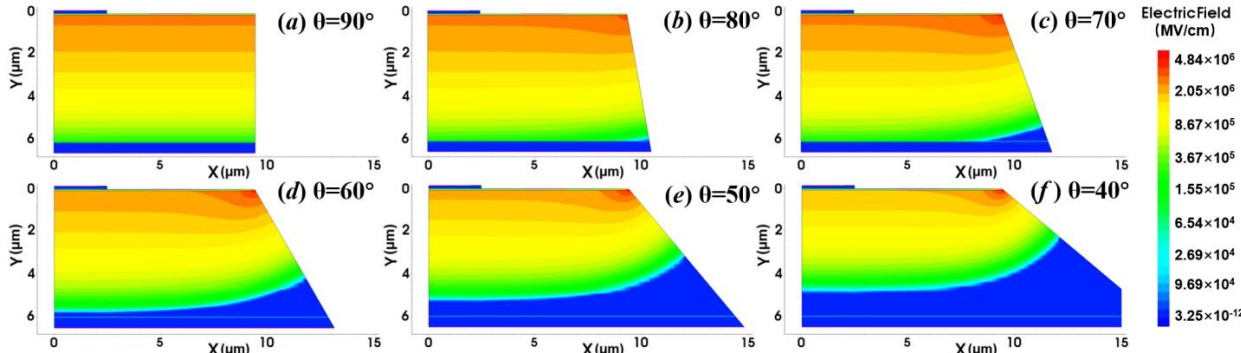

**Figure 10.** The 2D profile of electric field in the FV-GaN with θ = (**a**) 90°, (**b**) 80°, (**c**) 70°, (**d**) 60°, (**e**) 50°, and (**f**) 40°. It can be seen that as the beveled angle decreases, the electric field gradually concentrates at the edge of the BC mesa.

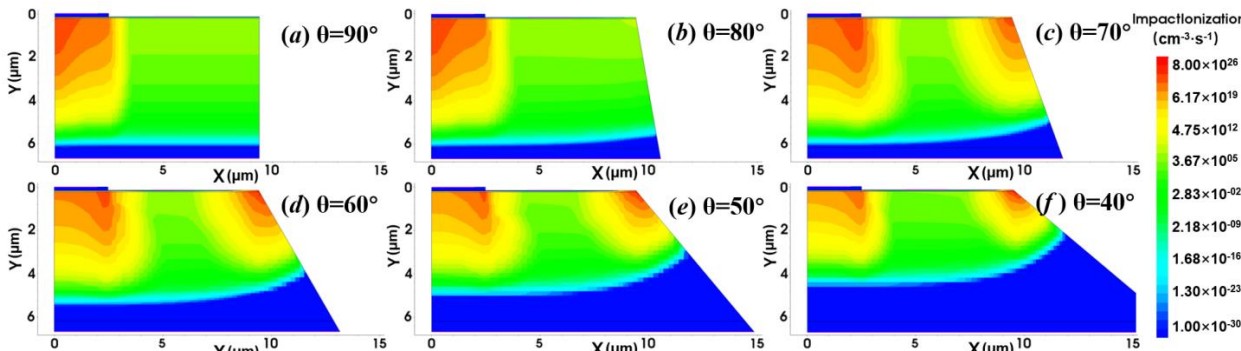

**Figure 11.** The 2D profile of I.I. in the FV-HBT with θ = (**a**) 90°, (**b**) 80°, (**c**) 70°, (**d**) 60°, (**e**) 50°, and (**f**) 40°. From the figure, it can be seen that as the beveled angle decreases, when avalanche breakdown occurs, a conductive channel appears on the BC mesa surface and the I.I. maximum gradually shifts from below the emitter (70°) to the edge of the BC mesa surface (60°).

According to the simulation, a breakdown voltage of 1270 V can be achieved with a doping concentration of $2 \times 10^{16}$ cm$^{-3}$ at the collector and a thickness of 6 μm, and the output curves and Gummel plots of FV-HBT with forward conduction are plotted based on this parameter for the two structures, both QV-HBT and FV-HBT (as shown in Figure 12). It is found that the highest saturation currents of 4.52 mA and 4.69 mA can be obtained at an $I_B$ of 100 μA with specific on-resistance $R_{on,sp}$ of 1.02 mΩ·cm$^2$ and 0.88 mΩ·cm$^2$, respectively. The FV-HBT shows a 13.7% improvement with respect to the QV-HBT due to having a more homogeneous current conduction path. The maximum DC gains of 66 and 73 can be achieved at $V_{BE}$ = 4.4 V and 4.9 V, respectively. For low-voltage HBTs, QV-HBTs are a good choice due to cost, as well as fabrication difficulty, but FV-HBTs are a better choice for high voltage and high power realization due to their more uniform current distribution, as well as the advantages of high current, high gain, and relatively less susceptibility to surface breakdown. The above simulation data provides design ideas for the fabrication of GaN-based HBTs with 1.2 kV and higher breakdown voltages.

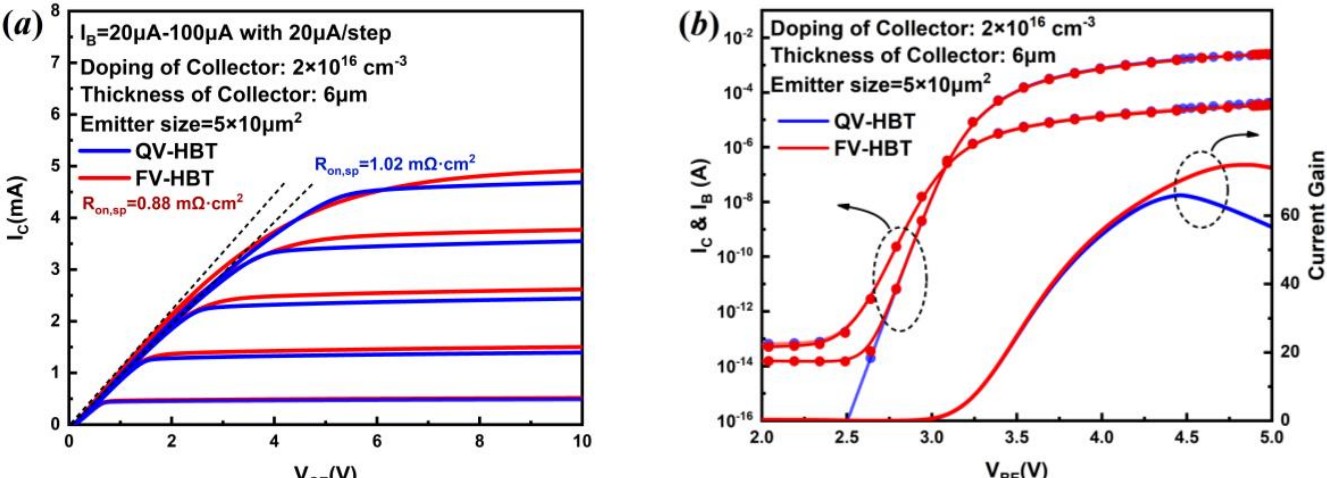

**Figure 12.** Simulated common-emitter (**a**) output characteristic curves; (**b**) Gummel plot of HBTs with a breakdown voltage of 1270 V.

In this paper, detailed and comprehensive modelling and calculation of 1.2 kV high-voltage GaN HBT is carried out for the first time, and it is concluded that the optimum Al fraction in AlGaN emitter is 7%, which is conducive to achieving higher current gain and improving power density. Secondly, the forward and reverse electrical characteristics of HBTs with different substrate types as well as different critical breakdown inclinations are

investigated, which provides a theoretical foundation for the design and the fabrication of GaN HBT power electronic devices. In addition, we introduce the relationship between different base/collector designs and GaN HBT breakdown, which is of great importance for the application of GaN HBTs in 1.2 kV, 3.0 kV, and higher voltage scenarios.

## 4. Conclusions

This study presents an analysis of gallium nitride (GaN) heterojunction bipolar transistors (HBTs) using TCAD. The focus is on investigating the impact of various key parameters on the gain characteristics, output characteristics, and breakdown characteristics of these devices. It has been observed that the DC gain of the AlGaN/GaN HBTs exhibits a nonlinear relationship with the Al fraction. By optimizing the concentration of the base and the concentration and thickness of the collector epitaxial layer, it is possible to achieve breakdown voltages of at least 1270 V. This is based on a collector thickness of 6 $\mu$m, a carrier concentration of $2 \times 10^{16}$ cm$^{-3}$, a specific on-resistance of 0.88 m$\Omega\cdot$cm$^2$, and a current gain of 73. Furthermore, we have conducted an investigation into the breakdown characteristics of two types of HBTs, namely QV-HBTs and FV-HBTs. This investigation involved varying the inclinations of the ramp and analyzing the resulting breakdown behavior. As a result, we have identified critical angles of 79° and 69° that should be adhered to in order to prevent surface breakdown of the device. This finding is significant as it contributes to the achievement of avalanches in GaN HBTs. It is anticipated that the aforementioned findings will offer valuable insights for informing the design process of GaN-based power HBTs in their subsequent development. This includes optimizing their performance in terms of high breakdown, high current density, and high power capabilities, thereby effectively harnessing the inherent advantages of GaN HBT devices.

**Author Contributions:** Conceptualization, X.W.; methodology, X.W. and L.Z.; formal analysis, L.Z., Z.C. and Z.L.; writing—original draft preparation, X.W.; writing—review and editing, J.H. and X.W.; supervision, Y.Z. All authors have read and agreed to the published version of the manuscript.

**Funding:** This work was supported in part by National Key Research and Development Program of China under Grant No. 2022YFB3604202 and in part by the CAS Project for Young Scientists in Basic Research under Grant YSBR-064. (Corresponding author: Yun Zhang.)

**Data Availability Statement:** The data that support the findings are available from the corresponding author upon reasonable request.

**Conflicts of Interest:** The authors declare no conflict of interest.

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
