# Peer review of "Simulation and Comprehensive Analysis of AlGaN/GaN HBT for High Voltage and High Current"

_electronics, doi:10.3390/electronics12173590_

Round 1

Reviewer 1 Report

The article "Simulation and Comprehensive Analysis of AlGaN/GaN HBT for High Voltage and High Current" by Xinyuan Wang et al. presents a comprehensive study of the impact of various key parameters on the gain characteristics, output characteristics, and breakdown characteristics of AlGaN/GaN HBTs. The authors have conducted a systematic study and their findings offer valuable insights for designing GaN-based power HBTs with elevated breakdown thresholds, heightened current densities, and increased power capabilities.

The paper is well-written, and the presentation is clear and concise with appropriate references. The authors have investigated the impact of various key parameters on the gain, output, and breakdown characteristics of these devices. The figures are well-labeled and easy to understand.

minor comments:

·         The authors should discuss the implications of their findings for the design of GaN-based power HBTs. For example, they could discuss how their results could be used to optimize the performance of these devices for specific applications.

·         The authors should clarify the meaning of the term "conventional avalanche" in the conclusions section.

These minor comments would improve the quality of the paper. However, I recommend that it be accepted for publication in electronics. The paper presents valuable insights into the design of AlGaN/GaN HBTs for high voltage and high current applications.

Reviewer 2 Report

In this manuscript, the authors analyzed gallium nitride HBTs using TCAD and investigated the impact of Al mole fraction, the concentration of the base and the concentration and thickness of the collector epitaxial layer, on the performance of the HBTs. It was found that 7% of the Al reaches the peak value. The breakdown voltage can be achieved to 1270 V. Additionally, authors proposed the critical angles of 80 and 60 for QV-HBTs and FV-HBTs to prevent surface breakdown.

The authors provide valuable insights for designing the GaN-based HBTs. The manuscript is generally well presented with sufficient data. One suggestion is that the authors can improve on the readability of the data by providing more detailed description in the captions. Readers should be able to under the data with the legends and captions of figures without referring to the context. For example, a scale bar should be provided for Figure 4(a), with a description of the lines a, b, a', b' in the captions. 

Some minor grammatical editing can help better readability.

Reviewer 3 Report

The authors propose simulation work in the context of 3D models of high performance GaN devices for HV and high power applications.

I would first like to thank the authors for submitting a well done and interesting work.

Here are some curiosities from me, and some comments to improve the exposition, since I consider the technical content already suitable for publication on MDPI Electronics.

I would advise the authors already in the motivations section to highlight why the focus is on this type of device and for what types of specific applications you think would benefit from a research work like yours. For example, if you have thought of Automotive-type applications for full electric vehicles where the current trend is to integrate high voltage power trains. Also strongly motivate the need to expand the state of the art on 3D modeling, comparing this work with works where on the contrary we try to reduce the computational complexity of the models. In this regard, I would recommend introducing a brief motivational speech by comparing today's works on less complex 2D models, highlighting the advantages (and obviously also the disadvantages). I recommend the following publications for this purpose:

https://www.mdpi.com/2079-9292/11/1/112

https://www.mdpi.com/1996-1073/12/19/3608

https://ietresearch.onlinelibrary.wiley.com/doi/full/10.1049/pel2.12527

https://ieeexplore.ieee.org/document/9242993

https://ieeexplore.ieee.org/abstract/document/10192428

A final discussion on the possibility of reusing the workflow, or even this model, to analyze SiC devices with equivalent characteristics would also be needed. This could have a very important practical implication in the design phases of power converters.

Certainly the computational complexity of 3D models is not a fundamental aspect, but a qualitative discussion of the complexity of the model, what kind of numerical solvers have been used, and what simulation times you have encountered would be welcome.

I hope my comments can be useful to you.

Congratulations for your work.
